



# Low-latitude climate change linked to high-latitude glaciation during the Late Paleozoic Ice Age: evidence from the terrigenous detrital kaolinite

Peixin Zhang[1], Jing Lu[1], Minfang Yang[2], Longyi Shao[1], Ziwei Wang[1], Jason Hilton[3]

[1]State Key Laboratory of Coal Resources and Safe Mining, College of Geoscience and Surveying Engineering, China University of Mining and Technology, Beijing 100083, PR China

[2]Research Institute of Petroleum Exploration and Development, PetroChina, Beijing 100083, PR China

[3]School of Geography, Earth and Environmental Sciences, The University of Birmingham, Edgbaston, Birmingham B15 2TT, UK

*Correspondence to*: Jason Hilton (j.m.hilton@bham.ac.uk) and Jing Lu (lujing@cumtb.edu.cn)

**Abstract.** The Late Paleozoic Ice Age (LPIA; *ca.* 360–260 million years ago) was one of the most significant glacial events in Earth history that records cycles of ice advance and retreat in southern high-latitude Gondwana and provides a deep-time perspective for climate-glaciation coevolution. However, climate records from the LIPA are poorly understood in low latitudes, particularly in the North China Plate (NCP) on the eastern Palaeo-Tethys. We address this through a

detailed mineralogical study of the marine-continental sedimentary succession in the Yuzhou Coalfield from the southern NCP in which we apply Zircon U-Pb dating, biostratigraphy, and high-resolution clay mineral composition to reconstruct latest Carboniferous to early Permian chronostratigraphy and climate change. The Benxi, Taiyuan, and Shanxi formations in the study area are assigned to the Gzhelian, Asselian-Artinskian, and Kungurian-Roadian stages respectively and the Carboniferous Permian lithostratigraphy across NCP recognized as widely diachronous. Detrital micromorphology of

kaolinite under scanning electron microscopy and illite crystallization indicates kaolinite contents to be a robust proxy for palaeoclimate reconstruction. Kaolinite data show alternating warm-humid and cool-humid climate conditions that are roughly consistent with the calibrated glacial-interglacial successions recognized in high-latitude eastern Australia, including the glaciations P1 (Asselian-early Sakmarian) and P2 (late Sakmarian-early Artinskian), as well as the climatic transition to glaciation P3 (Roadian). Our results indicate a comparatively cool-humid and warm-humid climate mode in

low-latitude NCP during glacial and interglacial periods, and this is a significant step toward connecting climate change in low-latitudes to high-latitude glaciation during the LPIA.

## 1 Introduction

The Cisuralian (early Permian) represents the peak of the Late Paleozoic Ice Age (LPIA; *ca.* 332–260 Ma) in which

significant shifts from glacial to interglacial conditions occurred in high-latitude Gondwana (e.g., Montañez et al., 2007; Fielding et al., 2008; Montañez and Poulsen, 2013; Garbelli et al., 2019; Richey et al., 2020). These glacial-interglacial events caused fluctuations in the global climate and are recorded by indirect proxies from mid- and low-latitudes in marine and continental deposits (e.g., Korte et al., 2005; Grossman et al., 2008; Davydov, 2014; Yang et al., 2016, 2020). However, previous studies on early Permian climate change links to high-latitude glaciation mainly concentrated on the



western Tethys and mid-continental Euramerica (e.g., Montañez et al., 2007; Grossman et al., 2008; DiMichele et al.,
2010, 2014; Davydov, 2014), whereas little is known in low-latitude continental settings of the North China Plate (NCP)
in the eastern Palaeo-Tethys.

Kaolinite content of mudrock is an important climate proxy and has been widely used to reconstruct palaeoclimatic
change over different timescales (Singer, 1984; Fring et al., 2019). Previous studies have demonstrated that kaolinite is a
product of weathering controlled by climatic factors (mainly temperature and humidity), with palaeoclimatic information
from them typically agreeing with conclusions from other proxies in marine and continental deposits including
palaeopalynology and oxygen-isotope analysis (Singer, 1984; Fring et al., 2019). Although clay mineral composition (e.g.,
kaolinite and illite) is an important climate proxy that has been successfully used to reconstructing past climate
conditions from eastern Palaeo-Tethys (e.g., Cheng et al., 2019; Lu et al., 2020), few studies have considered
palaeoclimates from clay mineral proxies linked to glacial events during the early Permian.

In this paper, we use two new radiometric dates from interbedded tuffaceous claystone horizons and revised
conodont biostratigraphy as well as previous fusulinid and plant megafossil biostratigraphy and lithostratigraphy (Wang
and Shang, 1989; Wang, 2003; Gao et al., 2005; Pei, 2004, 2009; Yang and Wang, 2012) to constrain the age of the
studied strata in the Yuzhou Coalfield of the southern NCP, enabling its correlation with well-dated glaciations. We use
the clay mineral kaolinite as a proxy to reconstruct palaeoclimate and to reveal the links between low-latitude climate
change and late Pennsylvanian-earliest Guadalupian high-latitude glaciation. This provides significant insights into
low-latitude signals for climate change during the LPIA.

## 2 Materials and methods

During the early Permian, the NCP was located at approximately 5–15°N on the northern margin of the Palaeo-Tethys
Ocean (PTO) (Liu, 1990; Blakey, 2011; Fig. 1a). Previous analysis of palaeocurrent and palaeogeography shows that
during the Permian the study area was higher in the north and lower in the south (Yang and Lei, 1987). Sediments were
mainly derived from the Inner Mongolia Uplift (IMU) in the northern part of the NCP following ongoing southerly
subduction of the Palaeo-Asian Ocean beneath the NCP (Yang and Lei, 1987; Liu, 1990; Fig. 1b). Lithologies in the IMU
source area mainly comprise Archean gneiss and granulite, followed by less frequent Archean granite and Proterozoic
sedimentary rocks (Zhou, 2002).

The lithostratigraphic units studied in this paper include the Benxi (= Penchi), Taiyuan, and Shanxi (= Shansi)
formations in ascending order. In the Henan Province of the southern NCP, the target strata unconformably overlie
Ordovician gray-black marine limestones and contact conformably with overlying Xiashihezi Formation (Yang and Lei,
1987; Yang, 2006). The Benxi Formation comprises three members in Henan Province (Pei, 2004, 2009; Fig. 1c), but
only the upper member is developed in the study area, and mainly comprises gray-white iron and aluminum-rich
mudrocks deposited (Yang, 2006; Pei, 2009; Fig. 1c). The Taiyuan Formation includes the Dajian, Mojie, and Zhutougou
members in ascending order, and comprise mainly terrigenous mudrock and marine limestone beds (numbered #$L_1$ to #$L_9$)
with one layer of sandstone (the Hushi sandstone) and some thin coal seams (Yang, 2006; Pei, 2009; Fig. 1c).

In the Shanxi Formation, the basal #$2_1$ Coal Member includes the economically important #$2_1$ coal seam and is
dominated mainly by grayish-black mudrocks (Yang, 2006; Pei, 2009; Fig. 1c). The Dazhan and Xiangtan Sandstone (DZ
& XT) Member (Fig. 1c) mainly encompasses gray, fine-medium sandstone, and gray mudrocks with thin coal seams.
The Dazhan and Xiangtan sandstones were deposited as distributary channels or river mouth bars in a deltaic setting,



both also constitute regional lithostratigraphic marker beds that crop through the Henan Province (Yang and Lei, 1987). At the top of the Shanxi Formation, the Xiaozi Mudrock Member (XZ M.) (Fig. 1c) comprises purplish-red and

grayish-green mudrock deposited in delta plain environments and also represent an important lithostratigraphic marker (Yang and Lei, 1987); coals are absent in this member (Fig. 1c).

From the ZK21-1 borehole (34°18′3″N, 113°20′13″E) in the Yuzhou Coalfield, fresh mudrock (50 samples), were collected from the Benxi to the Shanxi formations (sampling locations shown in Figure 4). Samples were ground to less than 200 mesh and divided into three subparts for (1) clay mineral analysis, (2) major elements analysis, and (3) trace

elements analysis. Clay mineral compositions analysis using an X-ray diffractometer (D/max 2500 PC) at the State Key Laboratory of Coal Resources and Safe Mining (Beijing), and the data were interpreted using Clayquan 2016 software with a relative analysis error of ±5%. Major elements analysis was undertaken using an X-ray fluorescence spectrometer (PW2404) at the Beijing Research Institute of Uranium Geology with the relative analysis error of ± 5%. Trace elements analysis was undertaken using an inductively coupled plasma mass spectrometer (Finnigan MAT) at the Beijing Research

Institute of Uranium Geology with the relative analysis error better than ± 5%. The micromorphology of kaolinite was observed under scanning electron microscopy (SEM, FEI Nova Nano SEM450) with an energy dispersive spectrometer (Oxford instrument X-Max[80] detector) at the Beijing Research Institute of Uranium Geology.

From the Dafengkou section (34°9′53″N, 113°10′57″E) in the Yuzhou Coalfield, tuffaceous claystone samples were collected from the top of the Benxi Formation (HYD-1) and the upper member of the Shanxi Formation (HYD-2) for

zircon U–Pb dating (sampling locations shown in Figures 2 and 4). The two tuffaceous claystone beds, each lacking bedding features and fossils, vary from 20 to 50cm in thickness and are gray-white, homogenous, and powdery claystone (Fig. 3a, b). After crushing, grinding, sieving, and heavy liquid and magnetic separation, euhedral zircon crystals with clear oscillatory zoning under cathodoluminescence (CL) microscope were selected for U-Pb zircon isotope analysis. U-Pb dating was conducted at the State Key Laboratory Geological Processes and Mineral Resources (Beijing), using a

Thermo Fisher's X-Series 2 ICP-MS instrument. More details of the analytical method follow those of Lu et al. (2021a, b and references therein). Nine fresh limestone samples were collected for conodont biostratigraphy from the Taiyuan Formation (sampling locations shown in Figure 4); it was not possible to collect from the #$L_6$ limestone because there were no exposed strata were observed in the field. Conodont samples were crushed to <5mm pieces and dissolved in a solution of 6–10% acetic acid that was buffered with tricalcium phosphate. The acid and buffer were exchanged every

48h until the samples were fully dissolved. Residual materials were wet sieved, dried at 50°C and separated for heavy fractions using sodium polytungstate. Conodont elements were picked under a binocular microscope at the State Key Laboratory Geological Processes and Mineral Resources (Beijing).

As the abundance of kaolinite in modern sediments is dependent on the intensity of chemical weathering controlled by climate, plus its strong resistance to diagenesis (Singer, 1984; Thiry, 2000), changes in terrigenous kaolinite content of

clay minerals in this study were used to reconstruct palaeoclimatic change of the target strata. At the same time, the influences of depositional recycling, hydraulic sorting, and post-deposition diagenesis on kaolinite content has been evaluated by the Th/U ratios, the correlation analysis of Al/Si ratios, K/Si ratios, and Weathering Index of Parker (WIP), and illite crystallinity (KI) and kaolinite micromorphology of under SEM (e.g., Chen et al., 2003; Bauluz et al., 2008; Roy and Rose, 2013; Yang et al., 2016, 2018; Cheng et al., 2019).



## 3  Results and analysis

### 3.1  U-Pb dating of zircon and conodont assemblages

More than 1500 zircon crystals were separated from the tuffaceous claystone sample HYD-1 and 1000 from tuffaceous claystone HYD-2, with crystal sizes varying from 50 to 200 μm (Fig. 3a). The results of zircon U–Pb dating are shown in Figure 3 and Table S1. From sample HYD-1, a total of 11 concordant age values of youngest zircon shows a weighted mean $^{206}Pb/^{238}U$ age of 299.4 ± 1.8 Ma (mean squared weighted deviation (MSWD) = 0.17, n = 11) (Fig. 3b). Sample HYD-2 yielded 5 concordant age values of youngest zircon shows a weighted mean $^{206}Pb/^{238}U$ age of 270.7 ± 2.0 Ma (MSWD = 2.8, n=5) (Fig. 3c).

Most crystals show euhedral morphology and clear oscillatory zoning in cathodoluminescence (CL) (Fig. 3a). The Th/U ratios of the zircon crystals vary from 0.42 to 1.81 (arithmetic mean ($\bar{x}$) = 1.01, Table S1). Collectively these features indicate that these are volcanic-sourced zircons (Yang et al, 2014; Lu et al., 2021a, b). Thus, we interpreted the weighted mean ages from the samples HYD-1 and HYD-2 as the sedimentary ages.

A total of 28 species of conodont have been identified in the studied strata (Figs. 4, 5; Table S3). Among them, the first occurrence of *Streptognathodus isolatus* in limestone #L$_1$ occurs at the base of the Asselian stage (Figs. 4, 5a-5c; Table S3) and defines the Carboniferous-Permian (C-P) boundary (Wang, 1991; Wang and Qi, 2003; Shen et al., 2019). The first occurrence of *Streptognathodus cristellaris* in limestone #L$_3$ (Figs. 4, 5d-5g; Table S3) and *Streptognathodus constrictus* in limestone #L$_4$ (Figs. 4, 5h-5j; Table S3) occur in the middle and upper part of the Asselian stage, indicating the #L$_3$ and #L$_4$ limestones were deposited during the Asselian stage of the Permian (Shen et al., 2019). The first occurrence of the conodont *Sweetognathus* aff. *Whitei* in limestone #L$_8$ (Figs. 4, 5k, 5l; Table S3) occurs at the base of the Artinskian stage of the Permian and defines the Sakmarian-Artinskian boundary (Shen et al., 2019).

### 3.2  Th/U, Al/Si and K/Si ratios, WIP, KI, and micromorphology of Kaolinite

The analysis result of Th/U, Al/Si, and K/Si ratios, WIP, KI, and micromorphology of Kaolinite are shown in Figure 4 and Tables S2, S3. Th/U ratios vary from 0.95 to 5.82 ($\bar{x}$ = 4.2) (Table S2), indicating little or no effect from depositional recycling on the samples. This is because recycled mudrocks exhibit high Th/U ratios of around 6 due to oxidation of $U^{4+}$ to $U^{6+}$ and its removal through soluble (e.g., Bhatia and Taylor, 1981; Lu et al., 2020). Combined with the stable uplift of provenance area (Liu, 1990) and the relative stability of parent rock and lithology (Zhou, 2002), we exclude the influence of re-cycle on kaolinite content.

Al/Si ratios vary from 0.21 to 0.91 ($\bar{x}$ = 0.43), K/Si ratios vary from 0.002 to 0.102 ($\bar{x}$ = 0.044), and WIP vary from 2.40 to 42.64 ($\bar{x}$ = 25.92). The Al/Si ratios poor correlations with K/Si ratios ($r^2$ = -0.005, P<0.01, n = 50) and WIP ($r^2$ = -0.049, P<0.01, n = 50) (Fig. S1, Table S3) indicate that clay mineral compositions as weathering activity products are not controlled by the hydraulic or sedimentary sorting process (e.g., Yang et al., 2016, 2018).

KI values vary from 0.22 to 1.88 Δ°/2θ ($\bar{x}$ = 0.49 Δ°/2θ), indicating that the mudrock samples were not affected by the diagenesis (e.g., Cheng et al., 2019). The micromorphology of kaolinite mostly presents irregular fragments with the size of 1–5 μm under the SEM (Fig. 6), which is different from the authigenic kaolinite in the morphology of vermiform and accordion formed during the diagenetic stage (e.g., Bauluz et al., 2008). This indicate the kaolinite of this study is of detrital origin, which is consistent with previous studies in the study area (Yang and Lei, 1987).

In sum, we conclude that the kaolinite in the studied strata is of terrigenous detrital origin, which is not obviously affected by the four factors above, and it is a reliable proxy for palaeoclimate reconstruction.



### 3.3 Clay mineral compositions

The analysis results of clay mineral compositions are shown in Figure 4 and Table S2. Clay minerals mainly consist
of illite-smectite mixed layers (0–65 %, x̄ = 41.7 %) and kaolinite (19–67 %, x̄ = 39.0 %), followed by illite (1–35 %, x̄ = 13.1 %) and chlorite (0–21 %, x̄ = 6.2 %) (Figs. 4, 7). Variations in kaolinite content allow the succession in this study to be divided into six stages in ascending order (S-I to S-VI; Fig. 4). Three lower value intervals of kaolinite content in the Benxi Formation (S-I; x̄ = 62.3%), the middle part of the Dajian to lower part of the Zhutougou members (S-III; x̄ = 53.3 %), and the upper part of the Zhutougou (Taiyuan Formation) to the DZ & XT (Shanxi Formation) members (S-V; x̄ = 40.9%) respectively (Fig. 4). Three lower value intervals of kaolinite content occur in the Dajian to lower part of the Mojie members (S-II; x̄ = 25.2 %), the upper part of the Dajian to the lower part of the Zhoutougou members (S-IV; x̄ = 27.7 %), and the Xiaozi Mudrock members (S-VI; x̄ = 22.0%) respectively (Fig. 4).

### 4    Discussion

#### 4.1   Stratigraphic correlation and division

Recent zircon dating of tuffs and tuffaceous claystones in the NCP shows Carboniferous-Permian (C-P) lithostratigraphic units are widely diachronous (Peng and Chen, 2003; Yang et al., 2020; Lu et al., 2021a; Wu et al., 2021; Fig. 8). In the middle of the NCP, the C-P boundary occurs in the middle of the Taiyuan Formation in the Palougou section of Baode county and the Shimenzhai section in the Liujiang Coalfield, but in the Wuda Coalfield the C-P boundary occurs near the top of the Taiyuan Formation (Schmitz et al., 2020; Lu et al., 2021a; Wu et al., 2021; Fig. 8). The Taiyuan Formation is more than 180 meters thick in Wuda but it much thinner in Baode (80m) and Liujiang (75m) (Fig. 8). Similarly, the Asselian stage in the Palougou section includes four lithostratigraphic units and comprises the upper part of the Taiyuan Formation to the Lower part of Shangshihezi Formation (Wu et al., 2021; Fig. 8), but at Yongcheng in Henan Province, only the upper part of Taiyuan Formation is assigned to the Asselian stage based on high-resolution Zircon dating (Yang et al., 2020; Fig. 8).

Detailed biostratigraphic studies have been previously been undertaken in the study area using conodonts, fusulinid and plant fossils (e.g., Wang and Shang, 1989; Wang, 2003; Pei, 2004, 2009; Gao et al., 2005; Yang and Wang, 2012). In this study in addition to biostratigraphy we obtained a zircon U-Pb age of 299.4 ± 1.8 Ma at the top of the Benxi Formation (HYD-1) that is very close to the age of the C-P boundary in the middle part of Taiyuan Formation in the Yongcheng Coalfield (299.32 ± 0.12 Ma) (Yang et al., 2020; Fig. 8). The first occurrence of the conodont *Streptognathodus isolatus* in this study (Figs. 4, 5) and the fusulinids *Sphaeroschwagerina* and *Pseudoschwagerina* in Limestone #L₁ (Yang and Lei, 1987; Wang and Shang, 1989; Wang, 2003, Pei, 2004; Gao et al., 2005), together constrain the position of the C-P boundary to the top of Benxi Formation in the Yuzhou Coalfield.

Zircon dating results in the Yongcheng Coalfield constrain the position of the Asselian-Sakmarian boundary to the top of the Taiyuan Formation and coincides with a change in conodonts from *Streptognathodus* to *Sweetognathus* (Gao et al., 2005; Yang et al, 2020; Fig. 8). Although we did not observe a change from *Streptognathodus* to *Sweetognathus* from the samples we prepared, previous studies have shown this occurs in the Limestone #L₄ in the Yuzhou Coalfield (Wang and Zhang, 1985; Ding and Wan, 1990; Wang and Qi, 2003; Pei, 2004, 2009; Fig. 4). Furthermore, the first occurrence of the conodonts *Streptognathodus cristellaris* in limestone #L₃ and *S. constrictus* in limestone #L₄ in this study (Figs. 4, 5) indicate that limestones #L₃ and #L₄ are of Asselian age (e.g., Shen et al., 2019) and place the base of the Sakmarian



stage above limestone #L$_4$ (Figs. 4, 9). In addition, the first occurrence of the conodont *Sweetognathus* aff. *Whitei* in
limestone #L$_8$ (Figs. 4, 5) allows us to place the base of the Artinskian stage to between limestones #L$_7$ and #L$_8$ (Shen et
al., 2019; Figs. 4, 9).

The Shanxi Formation was previously considered to be of Kungurian-Roadian age based on the presence of the
*Emplectopteris triangularis - Lobatannularia sinensis - Emplectoperidium alatum - Cathaysiopteris whitei* fossil plant
assemblage (Yang, 2006; Yang and Wang, 2012; Fig. 1c). In this study, a zircon U-Pb age of 270.7 ± 2.0 Ma from
sample HYD-2 in the Xiaozi Mudrock member is very close to the age of the Kungurian-Roadian boundary (Figs. 4, 9).
As a result, we roughly assign the Xiaozi Mudrock member to the Roadian stage and the remaining parts of the Shanxi
Formation to the Kungurian stage.

**4.2 Low-latitude NCP climate change linked to the high-latitude glaciation**

The relationship between low-latitude kaolinite content in the study area and the high-latitude glacial cycles established
by previous studies is shown in Figure 9. The three higher kaolinite content intervals from S-I, S-III, and S-V are roughly
isochronous with climate warming interglacial periods and are consistent with an increase of global atmospheric $p$CO$_2$
(Richey et al., 2020; Fig. 9f) and high surface seawater temperatures (Korte et al., 2005; Grossman et al., 2008; Fig. 9e).
The three intervals with lower kaolinite content from S-II, S-IV, and S-VI roughly correspond to climate cooling glacial
periods from glaciations P1 and P2 and the transition period to glaciation P3 in high latitudes. The intervals with lower
kaolinite content are also are isochronous with an interval of decreasing global atmospheric $p$CO$_2$ (Foster et al., 2017;
Richey et al., 2020; Fig. 9f) and surface seawater temperatures (Korte et al., 2005; Fig. 9e). These patterns suggest that
the alternation of climatic changes indicated by kaolinite contents in the study area reflect global scale climate change
and high-latitude glacial-interglacial alternations.

Our result shows that climate change is the main factor controlling kaolinite content changes in the terrestrial
Yuzhou Coalfield (see sections 4.2 and 4.3) affecting temperature and humidity conditions. Considering a wet,
peat-forming climate persisted in the NCP during the early Permian (e.g., Hilton and Cleal, 2007; Yang and Wang, 2012;
Yang et al., 2016), we ascribe the variation in kaolinite content in the study area to temperature changes and thus infer
humid-cool and humid-warm climate prevailing in the low-latitude NCP and relate them to glacial and interglacial
periods respectively.

Glacial-interglacial transitions are associated with especially large and rapid changes in climate conditions that can
be readily recorded by bulk sediment chemistry (e.g., Frings, 2019; Wang et al., 2020). In this study, kaolinite content
was characterized by rapid decrease at the onset and rapid increase at the end of S-II (Figs. 4, 9). However, the rapid
climate change over the glacial P1 transition was not reported in the change of clay minerals from the Xikou section of a
passage linking the Palaeo-Tethys Ocean (PTO) and Panthalassa due to lower sampling resolution (Cheng et al., 2019;
Fig. 1a).

The rapid decrease of kaolinite content near the C-P boundary indicates rapid climate cooling in the NCP (Fig. 4)
and is also recorded by the continental chemical index of alteration (CIA) in the Yongcheng Coalfield (Yang et al., 2020).
During the early Permian widespread ice sheets (including sea ice) accumulated in the Southern Hemisphere in mid- and
high-latitudes (Fielding et al., 2008; Mory et al., 2008; Holz et al., 2010). The early Permian build-up of ice in the
Southern Hemisphere has been attributed to the decrease of $p$CO$_2$ caused by the re-expansion of palaeotropical
rainforests during equatorial Cathaysia (Cleal and Thomas, 2005) and the rapid post-eruptional weathering of the vast
basaltic rocks of the Skagerrak-Centered broad large igneous province in tropical latitudes (Yang et al., 2020).



The rapid increase of kaolinite content during the mid-Sakmarian may indicate rapid climate warming in the NCP
(Fig. 4). Similar rapid climate change has been recorded by the CIA in the Karoo and Kalahari Basin sections of south
Gondwana where it is associated with the initiation of glacial retreat (Scheffler et al., 2006). This has also been recorded
by the Mineralogical Index of Alteration (MIA) and the CIA in the Khalaspir Basin in south Gondwana where it has been
interpreted as marking deglaciation of glacial P1 and the onset of glacial retreat (Roy and Rose, 2013). During this time,
extensive ablation of ice sheets occurred in multiple Gondwanan basins (Fielding et al., 2008; Holz et al., 2010; Frank et
al., 2015). This has been related to lowering of the equatorial Hercynian mountains that led to a re-establishment of thick
soils, while the assembly of Pangaea promoted arid conditions in continental interiors that were unfavorable for silicate
weathering, causing $p$CO$_2$ to rise to levels sufficient to terminate the glacial event (Goddéris et al., 2017).

We consider that the rapid climate changes observed in this study evidenced by clay mineral composition change
may correspond to the onset of the early Permian icehouse and permanent deglaciation during the early Sakmarian, and
that glaciation P1 experienced rapid expansion and contraction as indicated by the changing in kaolinite content patterns
in this study. Furthermore, a significant decrease of kaolinite content during the Roadian stage in the study area (Fig. 4)
has also been recorded in the Xikou Section and interpreted as a result of the glaciation P3 in high-latitude eastern
Australia (Cheng et al., 2019). However, the glaciation P3 lasted about 2 Myr and was confined to the late Wordian-early
Capitanian based on brachiopod shell-based $^{87}$Sr/$^{86}$Sr calibration (Garbelli et al., 2019). This suggests that the cooling
climate in the study area is related to the interglacial transition between the glaciations P2 and P3 and into the glaciation
P3 rather than the glaciation P3 in high-latitude eastern Australia (Shen et al., 2020).

## 5 Conclusions

(1) Two tuffaceous claystone yielding weighted mean $^{206}$Pb/$^{238}$U ages of 299.4 ± 1.8 Ma in the latest Gzhelian and
270.7 ± 2.0 Ma in the Roadian combined with conodont, fusilinid and plant fossil biostratigraphy and lithostratigraphy
provide a refined stratigraphic framework for the latest Carboniferous and early Permian in the Yuzhou Coalfield. This
allows us to assign the Benxi, Taiyuan, and Shanxi formations to the Gzhelian, Asselian-Artinskian, and
Kungurian-Roadian stages respectively, and demonstrates the diachronous nature of the Carboniferous and early Permian
lithostratigraphy in the NCP.

(2) X-ray diffraction analysis shows that clay mineral composition in the Yuzhou Coalfield mainly consists of
illite-smectite mixed layers ($\bar{x}$ = 41.7%) and kaolinite ($\bar{x}$ = 39.0%), followed by illite ($\bar{x}$ = 13.1%) and chlorite ($\bar{x}$ = 6.2%).
Detrital morphology of kaolinite under a scanning electron microscopy and illite crystallization indicates that kaolinite
values represent a robust proxy for palaeoclimate reconstruction and suggesting that variations in kaolinite values track
different prevailing climatic conditions.

(3) Kaolinite contents vary through the succession and present an alternating cool-humid and warm-humid climate
mode. This climatic mode at low latitudes recorded the glaciation P1 during the Asselian-early Sakmarian, the glaciation
P2 during the late Sakmarian-early Artinskian, and the glaciation and climatic transition to glaciation P3 during the
Roadian recorded in high-latitude eastern Australia.

(4) Two short-duration intervals with rapid climate change indicated by the changes in kaolinite content at the C-P
boundary and in the mid-Sakmarian correspond with the onset of the early Permian peak icehouse conditions and
permanent deglaciation respectively, implying that the glaciation P1 is characterized by a rapid expansion and rapid
contraction.



**Data and materials availability**

All data needed to evaluate the conclusions in the paper are present in the paper and/or the Supplementary Materials.

**Supplement**

Supplementary material for this article are available at [see Supplementary Information]

**Author contributions**

The study was designed by PXZ, JL, MFY, LYS and JH. Samples were collected by PXZ, JL, MY, and ZWW. Clay mineral, major and trace elements data acquisition and analyses were performed by JL, PXZ, MFY and JH, and other

analyses were performed by PXZ, JL, MFY and ZWW. PXZ, JL, and JH led the writing of the manuscript and discussed the results. All co-authors contributed to the interpretation of the data and to the final manuscript.

**Competing interests**

The authors declare no competing financial interests.

**Acknowledgments**

We are grateful to Fang Zhang for help in the identification of conodonts data, Suping Peng and Shifeng Dai (China University of Mining and Technology Beijing), Isabel Montañez and William DiMichele for comments on earlier versions of the manuscript, and Pengju Li and Linsong Liu for discussions on clay minerals.

**Financial support**

This study has been supported by the National Natural Science Foundation of China (Grant no. 41472131, 41772161),
the National Science and Technology Major Project (Award no. 2017ZX05009-002), and the New Century Excellent Talents Fund of Chinese Ministry of Education (Award no. 2013102050020).

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

**Figure captions**


**Figure 1.** Location and geological context for the study area. **a**, Palaeogeographic reconstruction for the Cisurlian (early
Permian) showing location of the North China Plate (NCP) (modified from Blakey, 2011); **b**, Palaeofacies map of the
NCP during the Cisuralian showing the location of study area (modified from Liu, 1990). **c**, Stratigraphic framework and
fossil distributions for the studied Carboniferous-Permian strata of the Benxi, Taiyuan, Shanxi and Xiashihezi (Xs)
formations from Henan Province in the southern NCP, with stratigraphic age revised from Shen (2019). Lithology





column derived from Pei (2004) and Yang (2006) with colors representing those of the strata in the field (section 2). $L_0^1$ to $L_9$ represent the position of individual limestone marker beds. Fossil plant assemblages (0–9) from Yang (2006) and Yang and Wang (2012). Fusulinid stratigraphic ranges and biozones from Pei (2004, 2009). Abbreviations: A. = Age; S. = Stages; Ord. = Ordovician; Kas. = Kasimovian; F. = Formation; M. = Member; $C2_1$# = Coal $2_1$ seam Member; DZ & XT

S. = Dazhan and Xiangtan Sandstone Member; XZ M. = Xiaozi Mudrock Member; Mid. = Middle; Up. = Upper; Lith. = Lithology; M.b. = Marker bed; HS= Hushi sandstone; $2_1$# = Coal $2_1$ seam; DZ = Dazhan sandstone; XT = Xiangtan sandstone; XZ = Xiaozi mudrock; SG. = Shaguoyao sandstone.

**Figure 2.** Photographs of the Dafengkou section (about 18km to the southwest of 21-1 borehole) in the Yuzhou Coalfield

showing features of the tuffaceous claystone deposit sampled. **a,** Boundary of the Benxi and Taiyuan formations with highlighted box enlarged in b. **b,** Enlargement from a showing details of the HYD-1 sample in the uppermost part of the Benxi Formation. **c**: Boundary of the Shanxi and Xiashihezi formations with highlighted box enlarged in d. **d,** Enlargement from c showing details of the HYD-2 sample in the uppermost Shanxi Formation. Abbreviations: Fm. = Formation; Sx F. = Shanxi Formation; Xs F. = Xiashihezi Formation; XZ M. = Xiaozi mudrock; SGY S. = Shaguoyao

sandstone.

**Figure 3.** Representative cathodoluminescence (CL) images, U-Pb probability density, and weighted mean ages diagrams for dated zircons from tuffaceous claystone horizons in the study area. **a,** Representative CL images of dated zircons, showing the sites of LA-ICP-MS U–Pb analyses. Blue circle = HYD-1 samples, red circle = HYD-2 samples, in each

case showing the position of zircon LA-ICP-MS U–Pb age-dating analysis, with numbers within the coloured circles showing sample numbers. **b,** U-Pb probability density and weighted mean ages diagrams for dated zircons from sample HYD-1, showing the youngest zircon weighted mean $^{206}Pb/^{238}U$ age of 299.4 ± 1.8 Ma (MSWD = 0.17, n=11, uncertainties are given at the 2σ level). **c,** U-Pb probability density and weighted mean ages diagrams for dated zircons from sample HYD-2, showing the youngest zircon weighted mean $^{206}Pb/^{238}U$ age of 270.7 ± 2.0 Ma (MSWD = 2.8, n=5,

uncertainties are given at the 2σ level).

**Figure 4.** Results from zircon U–Pb dating, conodont biostratigraphic ranges, clay mineral compositions, and illite crystallization from the Yuzhou Coalfield. Colors in the lithology column represent those seen in the field. Stratigraphic age revised from Shen et al. (2019). Bed numbers #$L_1$-$L_9$ refer to individual limestone horizons (Pei, 2004). Palaeocurrent

data from the Yuzhou Coalfield refer to those of Yang and Lei (1987) reveal that the palaeoflow flowed from north to south in the Shanxi Formation. Interpretation of deposition environments follows Liu (1987) and Yang and Lei (1987).



Abbreviations: Fm. = Formation; Ord. = Ordovician; Gz. = Gzhelian; C2$_1$# = Coal 2$_1$ seam Member; DZ & XT S. =

Dazhan and Xiangtan Sandstone Member; XZ M. = Xiaozi Mudrock Member; Dep. e. = Depositional environment; HS=

Hushi sandstone; #2$_1$ = Coal 2$_1$ seam; #2$_2$ = Coal 2$_2$ seam; DZ = Dazhan sandstone; XT = Xiangtan sandstone; XZ =

Xiaozi mudrock; SG. = Shaguoyao sandstone; Sam. = Sample number; I/S = illite-smectite mixed layers; I+C = illite +

chlorite; Illite cry. = illite crystallinity.

**Figure 5.** SEM images of example conodonts from limestones #L$_1$, #L$_2$, #L$_3$, #L$_4$ and #L$_8$ in the study area (all scale bars
= 100μm). **a-c**, *Streptognathodus isolatus* (#L$_1$ and #L$_2$); **d**, *Streptognathodus wabaunsensis* (#L$_1$); e, *Streptognathodus*
*elegantulus* (#L$_1$); **f**, *Streptognathodus gracilis* (#L$_1$); **g**, *Streptognathodus oppletus* (#L$_1$); **h**, *Streptognathodus parvus*
(#L$_1$); **i**, *Streptognathodus ximplex* (#L$_2$); **j**, *Streptognathodus nodulinearis* (#L$_2$); **k**, *Hindeodus minutus* (#L$_1$); **l**,
*Streptognathodus elongates* (#L$_2$); **m-p**, *Streptognathodus cristellaris* (#L$_3$); **q** and **r**, *Streptognathodus constrictus* (#L$_4$);
**s**, *Streptognathodus* cf. *constrictus* (#L$_4$); **t** and **u**, *Sweetognathus* aff. *whitei* (#L$_8$); **v**, *Sweetognathus inornatus* (#L$_8$).

**Figure 6.** SEM images and spectra of the amorphous component of kaolinite from mudrock in the study area. **a** and **d**,
SEM images of mudrock sample #348, #353, #363, and #375 showing the kaolinite in the studied strata are mostly
irregular fragments with the size of 1-5 μm, which is obviously different from the authigenic kaolinite occurs as
vermiform and accordion in diagenetic stage, representing the detrital origin of kaolinite in the studied strata; **e** and **h**,
spectra of the amorphous component of the kaolinite from sample #348, #353, #363, and #375.

**Figure 7.** X-ray diffraction (XRD) patterns of clay fractions of typical samples in the study area. N, E, and T designate
spectra of a naturally-oriented slide (red line), ethylene-glycol saturated for oriented slide (blue line), and
high-temperature treated (green line) at 450℃ for oriented slide, respectively. Abbreviations: I/S = illite-smectite mixed
layers; K+C = kaolinite + chlorite; I = illite; K = kaolinite; C = chlorite.

**Figure 8.** Chronostratigraphic and lithostratigraphic correlation of Carboniferous and Permian in the NCP. Lithology and
age data of the sections at Wuda (Zhou et al., 2015 and Schmitz et al., 2020), Baode (Wu et al., 2021), Liujiang (Lu et al.,
2021 and Unpublished data), Yongcheng (Yang et al., 2020), and Yuzhou (this study) organized from north to south on
the NCP during the Late Pennsylvanian to the earliest Guadalupian.

**Figure 9.** Comparison of climate change, glaciations, δ$^{18}$O, and $p$CO$_2$ records from the Gzhelian Stage of the
Carboniferous to the Roadian Stage of the Permian. Age stratigraphic framework from Shen et al. (2019); **a,** Climate
stages interpreted from the studied section; **b,** Interpreted climatic records with blue representing intervals of climate
cooling while yellow represents climatic warming based on fluctuating climate signals from kaolinite content from the
studied section, and climate type interpreted from the studied section. **c,** Records of glaciation events in Australia from
Fielding et al. (2008), Frank et al. (2015), and Garbelli et al. (2019); **d,** Oxygen isotope values from Korte et al. (2005)
and Grossman et al. (2008); **e,** Global $p$CO$_2$ concentration from Foster et al. (2017) and Richey et al. (2020). Note: the
red line from Foster et al. (2017), the black line comes from Richey et al. (2020). Abbreviations: Gzh. = Gzhelian; Roa. =



Roadian; Sak. = Sakmarian; Cli. S. = Climate stage; Clim. type = Climate type; $2_1$# = Coal $2_1$ seam; Temp. =

Temperature.





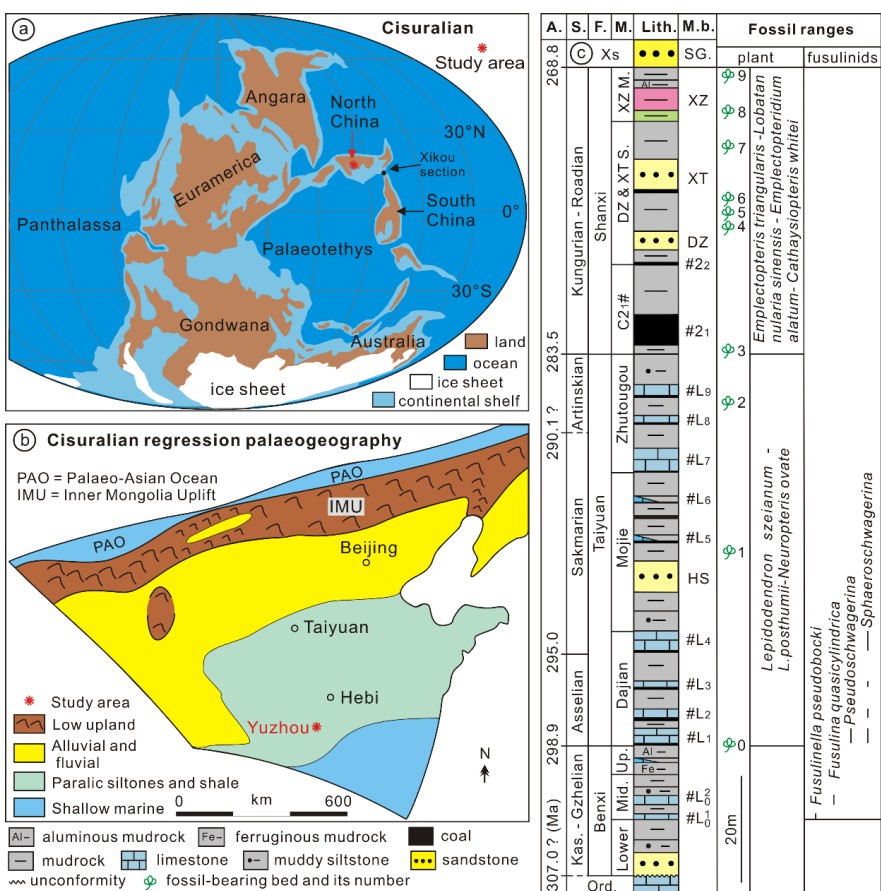

Figure 1


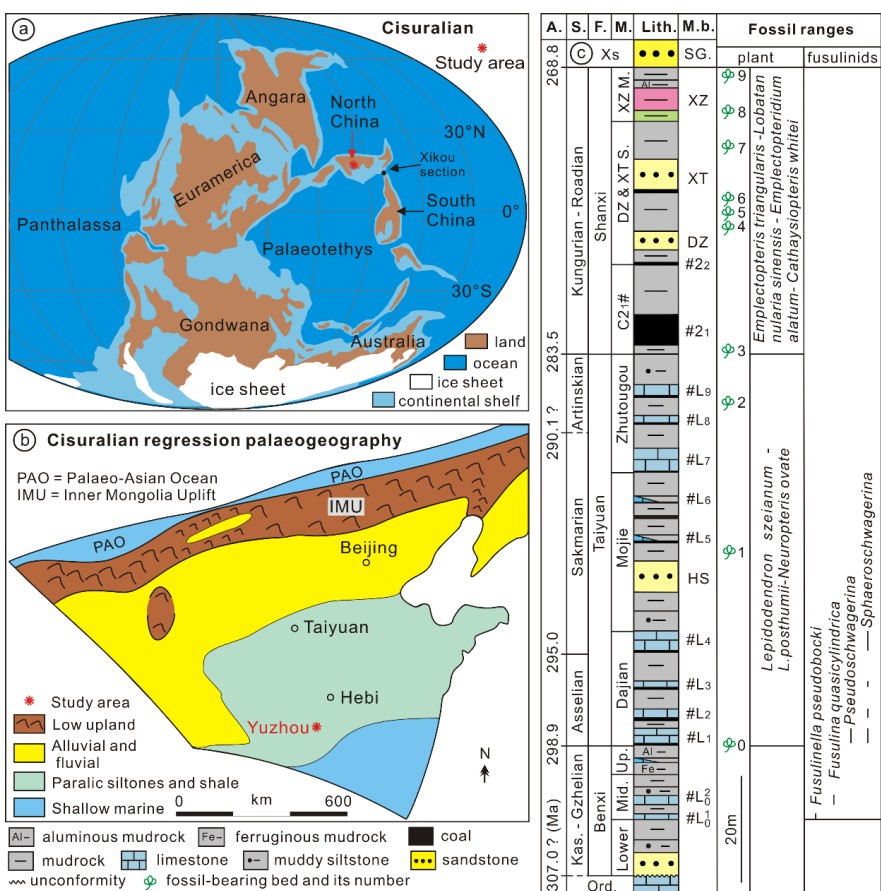

Figure 2

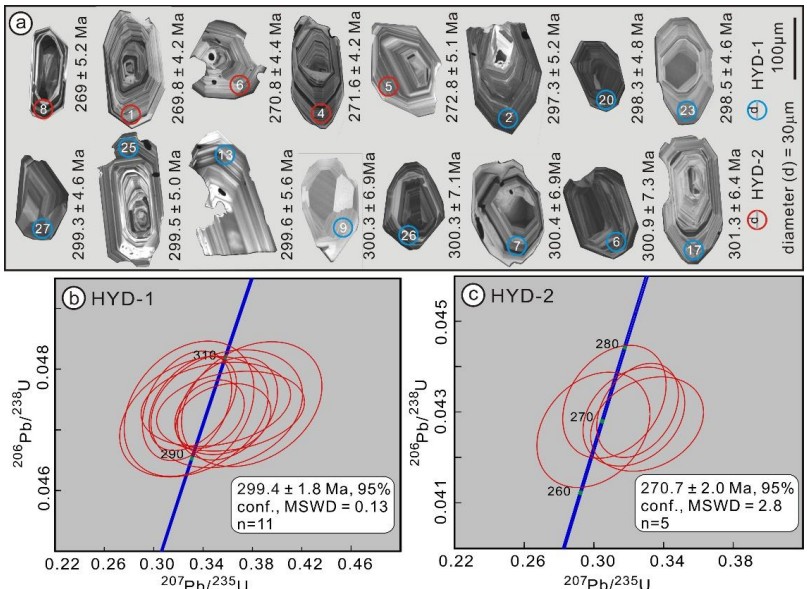

515                                   Figure 3

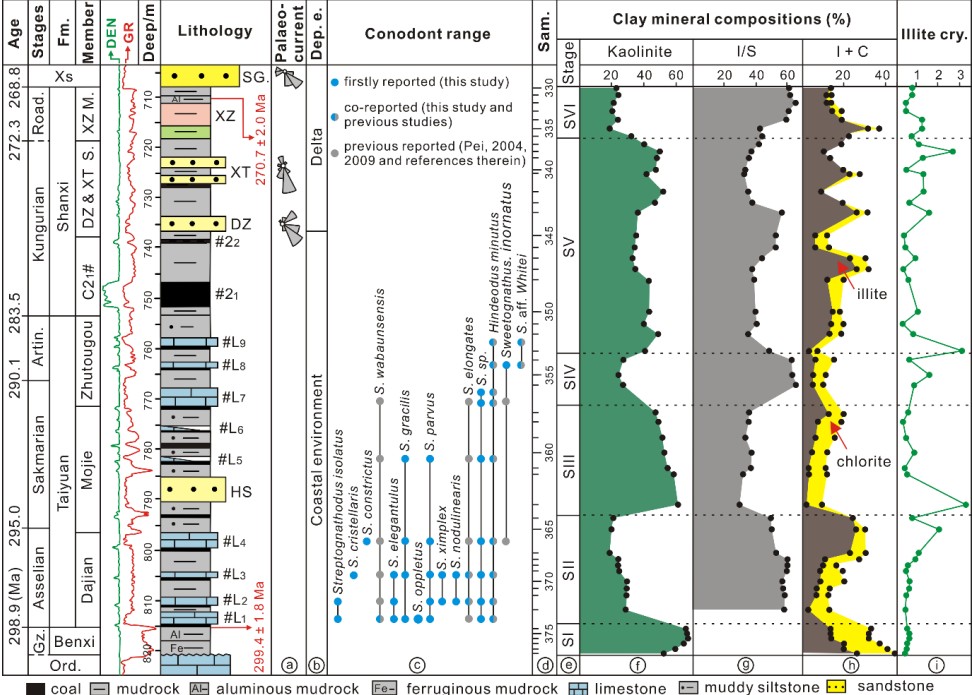

Figure 4





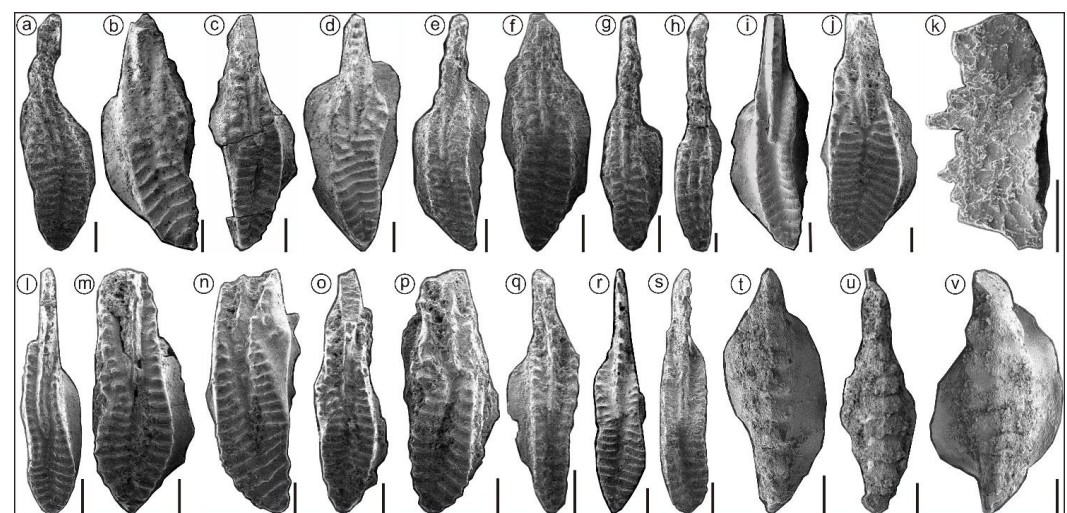


Figure 5

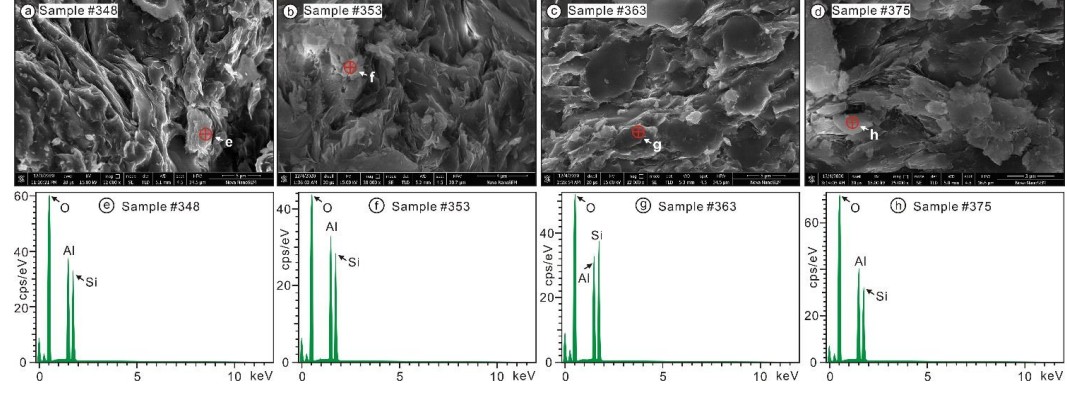


Figure 6



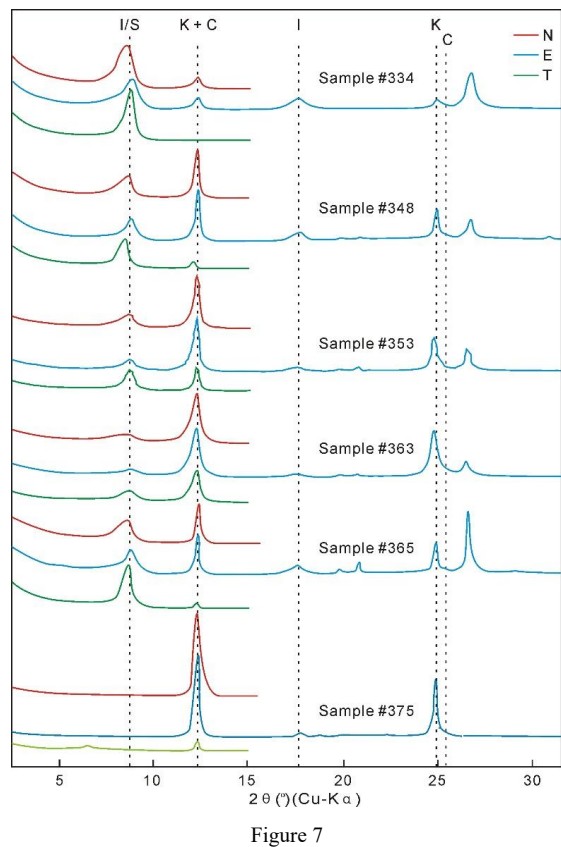

Figure 7



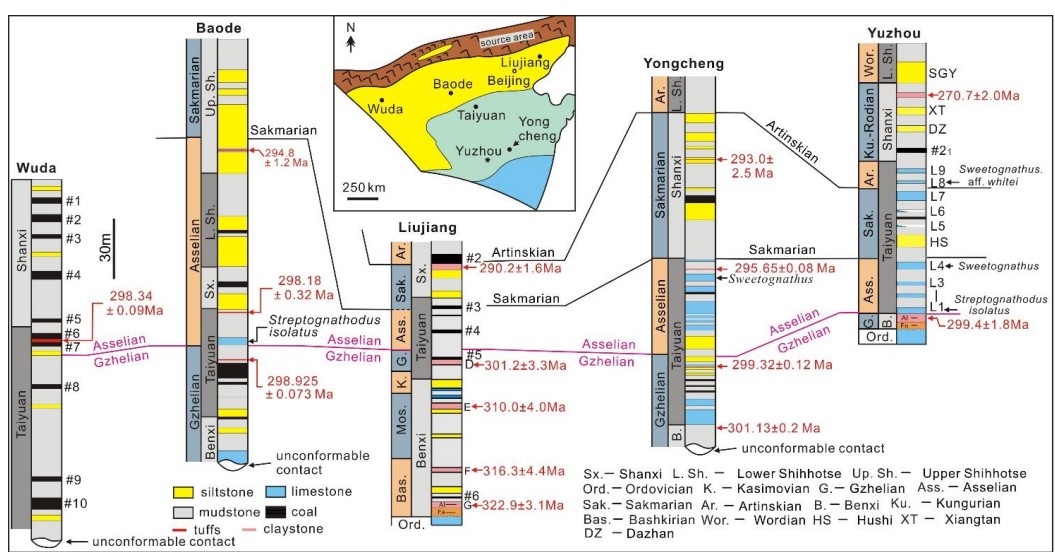

Figure 8

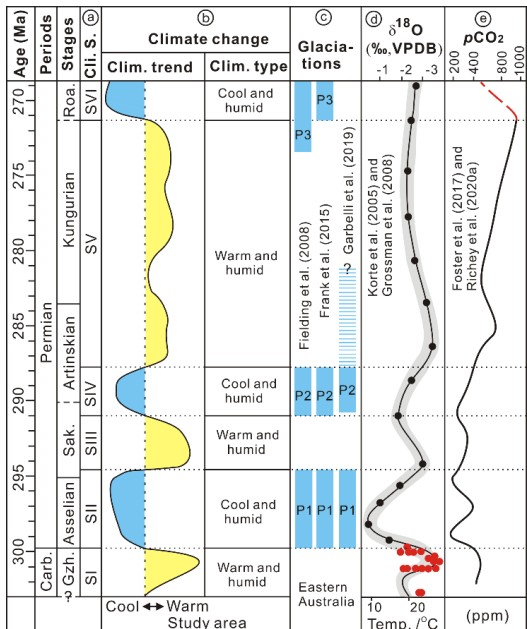

Figure 9