# Peer review of "Low-latitude climate change linked to high-latitude glaciation during the Late Paleozoic Ice Age: evidence from the terrigenous detrital kaolinite"

_Climate of the Past, 2021_

## Author Comment (AC1)

Dear reviewer,

Thank you for your assessment of our paper "**Low-latitude climate change linked to high-latitude glaciation during the Late Paleozoic Ice Age: evidence from the terrigenous detrital kaolinite**". The points raised are valid so the review provides us with the opportunity to expand our manuscript to address them comprehensively. However, we disagree with the recommendation of rejecting the manuscript because of the points raised as (i) the information is readily available and has now been incorporated into the revised manuscript, and (ii) this does not alter any of our results or conclusions in any way. Below we address each point individually and consider them fully resolved.

In the submission system, we cannot upload the revised manuscript. Therefore, we outline the changes we have made as indicated on the updated version of the manuscript showing marked changes (see below).

We hope you find these changes agreeable and we look forwards to hearing from you in the future.

Yours sincerely and on behalf of co-authors,

Jason Hilton

**Reviewer #1:**

**1.** The paper is well written and presents interesting results concerning past climate conditions for North China. However I identified a fundamental issue requiring clarification. Indeed the authors investigate local climate conditions **without presenting and discussing the geological setting of the area**. In the absence of data highlighting the geological evolution of this site, most of statements presented in the discussion (4.2) remain too speculative to support authors' conclusions. I encourage the authors to resubmit their article after significant rewriting of the manuscript, with a full description of the continental environment and its temporal evolution (which explains why I recommend "rejected" rather than "major revision").

**Response:** We thank the reviewer for their comments here. We added the contents about the geological background and environmental evolution of the study area in section 4.2, and we have added additional figures to demonstrate this.

The details are as follows: "Changes in continental plate position, land-sea distributions, ice extent in Gondwana, atmospheric $CO_2$ and monsoonal rainfall are considered the main factors controlling climate change in low latitudes during the LPIA (Tabor and Poulsen, 2008; Cao et al., 2017). In the study area, local paleomagnetic data demonstrate that no large-scale plate motion occurred from the early to late Early Permian (Zhu et al., 1996). Through this time interval, the region remained in an equatorial humid climate zone (Zhu et al., 1996) on the southern margin of the North China Plate (NCP) in proximity to the sea (Fig. 9). Following regression of the epicontinental sea in the earliest Permian (Fig. 9a, b),

sedimentary environments gradually developed from coastal tidal flat deposition to a series of shallow-water delta sedimentary systems comprising fluvial and tidally influenced offshore peat-bearing deposits (Zhu et al., 1996; Yang and Lei, 1987; Fig. 9). Fossil plants were widespread and abundant, comprising stable, tropical, ever-wet communities dominated by lycophytes, equisetophytes, marattialean ferns and pteridosperms (Zhu et al. 1996; Yang and Lei, 1987; Yang, 2006; Hilton and Cleal, 2007). Collectively, this information builds a picture of stability in continental plate position and land-sea distributions. The sedimentary record provides no evidence of monsoons. Consequently, we consider that waxing and waning of ice sheets in high latitude Gondwana and changes in global atmospheric $CO_2$ were the main factors affecting climate change in the study area through the LPIA."

[Figure]

**Figure 9**. Lithofacies palaeogeograpy map from late Bashkirian to Wordian in the North China Plate (modified from Shao et al., 2014).

**2.** One major issue is the determination of the paleolatitude of the site through time with good precision. In principle, local paleomagnetic data could decide this issue because from them paleolatitude can be estimated. Moreover, authors used a unique reconstruction (Blakey 2011) for a period extending over 30 myr (300-270Ma) without discussing most recent reconstructions with finer time slices (for details, see https://www.earthbyte.org/category/resources/data-models/paleogeography/). Because

the hydrological cycle (thus the weathering) depends on the Hadley cell in low latitudes this point must be solved in order to discuss the accuracy of findings in term of Earth's climate.

**Response:** We thank the reviewer for prompting us to scrutinize this information. According to the collected local paleomagnetic data (Table 1; see below), we determined the exact location of the study area through time and have modified this in the geological background section (see response 1). We have revised figure 1 and replaced "Blake's paleogeographic map" with the "generalized tectonic map of present-day China". In addition, we collected paleogeographic slices with higher time scale according to the suggestions of the reviewers (http://www.earthbyte.org/paleodem-resourcescotese-and-wright-2018/), and verified the accuracy of paleoclimate in the study area by further analyzing the changes of Hadley circulation during glacial interglacial period.

We collected palaeomagnetic data from the Dafengkou section in Yuzhou coalfield from early to late Early Permian. These data show that from the early to late Early Permian, the paleolatitude change in the study area is between 11.0°N and 11.4°N, that is, it is within the equatorial humid climate zone (Zhu et al., 1996).

Table 1 Permian ancient geomagnetic parameters and latitude in Dafengkou section

| Period | sampling point | mean direction of magnetization | | | α95 | stability valuation | Paleomagnetic pole position | | | | Paleolatitude position |
|---|---|---|---|---|---|---|---|---|---|---|---|
| | | D (°) | I (°) | K (°) | | | Plat.(°N) | Plong.(°N) | Dp | Dm | |
| $P^2_1$ | 6 | 143.2 | -21.2 | 12.9 | 19.4 | R | -49.2 | 177.4 | 10.7 | 20.4 | 11.0 |
| $P^1_1$ | 3 | 124.9 | -22.0 | 82.1 | 13.7 | R | -35.1 | 192.5 | 7.7 | 14.5 | 11.4 |
| $P_1$ | 9 | 136.9 | -21.7 | 16.1 | 13.2 | | -44.6 | 183.4 | 7.4 | 14.0 | 11.2 |

We have modified the paper follows: "Result from our study show that climate change, especially changes in temperature and humidity, is the main factor controlling kaolinite content changes in the terrestrial Yuzhou Coalfield (see sections 3.2 and 3.3). Evaluating detailed global palaeogeographic maps from 305.3 to 268.2 Ma superimposed with Hadley circulation patterns for glacial maximum and minimum conditions (Tabor and Poulsen, 2008; Fig. 10) provides further insights into the effects of climate change on the study area. According to results from palaeoclimate simulations (Tabor and Poulsen, 2008), the study area is affected by ITCZ compression during glacial P1 (Fig. 10c) and P2 (Fig. 10d) and the transition period of glacial P3 (Fig. 10g), leading to increased annual precipitation and weathering rates. The study area is affected by seasonal ITCZ expansion from the Kasimovian (Fig. 10a) to Gzhelian stage (Fig. 10 b), P1 (Fig. 10c) to P2 (Fig. 10d) interglacial stages and the P2 to P3 interglacial stage (Figs. 10e-f), decreasing annual precipitation and weathering rates. However, in our study, the change of weathering rates indicated by kaolinite content is inversely related to the change of annual precipitation in low latitudes during the glacial and interglacial periods (Fig. 11). Given that the study area was in the equatorial zone and under humid climates with abundant peat formation through the early Permian (e.g., Hilton and Cleal, 2007; Yang and Wang, 2012; Yang et al., 2016), we consider that the change of kaolinite content in the study area is mainly affected by the change of global temperature. This indicates relatively cool-humid and warm-humid climate prevailed in the low-latitude

NCP linked to glacial and interglacial periods respectively (Fig. 11)."

[Figure]

**Figure 10.** Global palaeogeographic maps from the Kasimovian stage of the Carboniferous to the Wordian stage of the Permian superimposed by Hadley circulation patterns for glacial maximum and glacial minimum conditions (modified from Tabor and Poulsen, 2008). Base maps modified from Scotese and Wright (2018) (http://www.earthbyte.org/paleodem-resourcescotese-and-wright-2018/). Abbreviations: R. and W. (P3 t.) = Roadian and Wordian (Glacier P3 transition); NC = North China; ITCZ = Intertropical Convergence Zone.

**3.** lines 195-200: the time-correlation deserves more attention. Based on the fig.9, the atmospheric $p$CO$_2$ for S-II and S-IV overcomes the one for S-3 while this state appears as "warm and humid".

**Response:** We thank the reviewer for their comments here. According to the time-correlation, we re-adjusted Figure 9 (now Figure 11) and added the CO$_2$ curve of Montañez et al. (2007). In addition, we revised the description of paleoclimate in the text.

According to the paleoclimate simulation results (Tabor and Poulsen, 2008), the study area is affected by seasonal ITCZ expansion during the P1 to P2 (Fig. 10d) interglacial stage, which will decrease of annual precipitation and further lead to the decrease of weathering rates. However, in our study, the change of weathering rates indicated by kaolinite content is inversely related to the change of annual precipitation in low latitudes during this interglacial period (Fig. 11). Considering that the study area was in the equatorial zone and the humid climate period of coal formation during the early Permian (e.g., Hilton and Cleal, 2007; Yang and Wang, 2012; Yang et al., 2016), we consider that the change of kaolinite content in the study area is mainly affected by the change of global temperature. Therefore, we consider that it is reasonable to interpret the climate in the study area as a relatively warm and humid climate during this time.

We have revised the manuscript to accommodate this as follows: "The rapid increase of kaolinite content during the mid-Sakmarian may indicate rapid climate warming in the NCP

(Fig. 4). During the middle Sakmarian, similar warm climate change has been recorded by the CIA in the Karoo and Kalahari Basin sections of south Gondwana where it is associated with the initiation of glacial retreat (Scheffler et al., 2006). This has also been recorded by the Mineralogical Index of Alteration (MIA) and the CIA in the Khalaspir Basin in south Gondwana where it has been interpreted as marking deglaciation of glacial P1 and the onset of glacial retreat (Roy and Rose, 2013). Evidence supporting the large-scale loss of the ice centers in high latitude Gondwana include: (1) marine strata in low latitudes generally indicating an obvious transgression during the middle and late Sakmarian as sea levels rose (Montañez et al., 2007; Stemmerik, 2008; Koch and Frank, 2011); (2) development of arid and semi-arid climates in the western Pangea (Poulsen et al., 2007; Montañez and Poulsen, 2013); (3) reduction in the extent and diversity of cold-water brachiopod faunas as temperatures rose (Clapham and James, 2008; Waterhouse and Shi, 2010); and (4) latitudinal gradients in global brachiopod biodiversity returned to comparable pre-LPIA levels (Powell, 2007). Although it is widely agreed that large-scale loss of several ice centers in high latitude Gondwana occurred during the mid-Sakmarian (Fielding et al., 2008; Holz et al., 2010; Frank et al., 2015; Richey et al., 2020), there is disagreement between global atmospheric $CO_2$ at this time increasing significantly (1000-2500ppm) (Montañez et al., 2007; Fig. 11) or experiencing a more limited rise (300-400ppm) (Richey et al., 2020; Fig. 11). Collectively these suggest that the ablation of the ice sheet $CO_2$ threshold was even lower than modeled (560 ppm; Lowry et al. 2014) during the earliest Permian (Richey et al., 2020). "

[Figure]

**Figure 11.** Comparison of climate change, glaciations, $\delta^{18}O$, and $pCO_2$ records from the Gzhelian Stage of the Carboniferous to the Roadian Stage of the Permian. Age stratigraphic framework from Shen et al. (2019); **a,** Climate stages interpreted from the studied section; **b,** Interpreted climatic records with blue representing intervals of climate cooling while yellow represents climatic warming based on fluctuating climate signals from kaolinite content from the studied section, and climate type interpreted from the studied section. **c,** Records

of glaciation events in Australia from Fielding et al. (2008), Frank et al. (2015), and Garbelli et al. (2019); **d,** Oxygen isotope values from Korte et al. (2005) and Grossman et al. (2008); **e,** Global $p$CO$_2$ concentration from Montañez et al. (2007), Foster et al. (2017) and Richey et al. (2020). Note: the red line from Foster et al. (2017), the black line comes from Richey et al. (2020). Abbreviations: Gzh. = Gzhelian; Sak. = Sakmarian; Cli. S. = Climate stage; Clim. type = Climate type; $2_1$# = Coal $2_1$ seam; Temp. = Temperature.

References:

Cao, W., Williams, S., Flament, N., Zahirovic, S., Scotese, C. and Muller, R. D.: Palaeolatitudinal distribution of lithologic indicators of climate in a palaeogeographic framework, Geol. Mag., 156(2), 331–354, doi:10.1017/S0016756818000110, 2019.

Cecil, C. B., Dulong, F. T., West, R. R., Stamm, R., Wardlaw, B. and Edgar, N. T.: Cliamte controls on the stratigraphy of a Middle Pennsylvanian cyclothem in North America, in Climate Controls on Stratigraphy, pp. 151–180, SEPM (Society for Sedimentary Geology)., 2003.

Clapham, M. E. and James, N. P.: Paleoecology Of Early-Middle Permian Marine Communities In Eastern Australia: Response To Global Climate Change In the Aftermath Of the Late Paleozoic Ice Age, Palaios, 23(11), 738–750, doi:10.2110/palo.2008.p08-022r, 2008.

Frank, T. D., Shultis, A. I. and Fielding, C. R.: Acme and demise of the late Palaeozoic ice age: A view from the southeastern margin of Gondwana, Palaeogeogr. Palaeoclimatol. Palaeoecol., 418, 176–192, doi:10.1016/j.palaeo.2014.11.016, 2015.

Koch, J. T. and Frank, T. D.: The Pennsylvanian-Permian transition in the low-latitude carbonate record and the onset of major Gondwanan glaciation, Palaeogeogr. Palaeoclimatol. Palaeoecol., 308(3–4), 362–372, doi:10.1016/j.palaeo.2011.05.041, 2011.

Lowry, D. P., Poulsen, C. J., Horton, D. E., Torsvik, T. H. and Pollard, D.: Thresholds for Paleozoic ice sheet initiation, Geology, 42(7), 627–630, 2014.

Montañez, I. P. and Poulsen, C. J.: The Late Paleozoic Ice Age: An Evolving Paradigm, Annu. Rev. Earth Planet. Sci., 41(1), 629–656, doi:10.1146/annurev.earth.031208.100118, 2013.

Montañez, I. P., Tabor, N. J., Niemeier, D., DiMichele, W. A., Frank, T. D., Fielding, C. R., Isbell, J. L., Birgenheier, L. P. and Rygel, M. C.: CO$_2$-Forced Climate and Vegetation Instability During Late Paleozoic Deglaciation, Science (80-. )., 315(5808), 87–91, doi:10.1126/science.1134207, 2007.

Poulsen, C. J., Pollard, D., Montañez, I. P. and Rowley, D.: Late Paleozoic tropical climate response to Gondwanan deglaciation, Geology, 35(9), 771–774, doi:10.1130/G23841A.1, 2007.

Powell, M. G.: Latitudinal diversity gradients for brachiopod genera during late Palaeozoic time: links between climate, biogeography and evolutionary rates, Glob. Ecol. Biogeogr., 16(4), 519–528, doi:10.1111/j.1466-8238.2007.00300.x, 2007.

Richey, J. D., Montañez, I. P., Goddéris, Y., Looy, C. V, Griffis, N. P. and DiMichele, W. A.: Influence of temporally varying weatherability on CO$_2$-climate coupling and ecosystem change in the late Paleozoic, Clim. Past, 16(5), 1759–1775, doi:10.5194/cp-16-1759-2020, 2020.

Roy, D. K. and Roser, B. P.: Climatic control on the composition of Carboniferous-Permian Gondwana sediments, Khalaspir basin, Bangladesh. Gondwana Res., 23, 1163–1171. doi: 10.1016/j.gr.2012.07.006, 2013.

Stemmerik, L.: Influence of late Paleozoic Gondwana glaciations on the depositional evolution of the northern Pangean shelf, North Greenland, Svalbard, and the Barents Sea, Resolv. Late Paleoz. Ice Age Time Sp., 441, 205, 2008.

Tabor, N. J. and Poulsen, C. J.: Palaeoclimate across the Late Pennsylvanian-Early Permian tropical palaeolatitudes: A review of climate indicators, their distribution, and relation to palaeophysiographic climate factors, Palaeogeogr. Palaeoclimatol. Palaeoecol., 268(3–4), 293–310, doi:10.1016/j.palaeo.2008.03.052, 2008.

Waterhouse, J. B. and Shi, G. R.: Evolution in a cold climate, Palaeogeogr. Palaeoclimatol. Palaeoecol., 298(1–2), 17–30, doi:10.1016/j.palaeo.2010.08.022, 2010.

Yang, G.: The Permian cathaysian flora in western Henan province, China-Yuzhou flora., 1–76, 2006.

Yang, Q. and Lei, S.: Depositional Environments and Coal-forming Characteristics of Late Palaeozoic Coal Measures in Yuxian, Henan Province., 1987.

Zhu, H., Yang, G. and Sheng, A. X.: A study on palaeomagnetism of Permian strata in the Dafengkou section, Yuzhou, Henan Province, Acta Geol. Sin., 70(2), 121–128, 1996.

---

## Author Comment (AC3)

Dear Editor,

Thank you very much for reviewing our paper "Low-latitude climate change linked to highlatitude glaciation during the Late Palaeozoic Ice Age: evidence from the terrigenous detrital kaolinite" and giving very valuable comments. The points raised are valid so the review provides us with the opportunity to expand our manuscript to address them comprehensively. Below we address each point individually and consider them fully resolved.

In the submission system, we cannot upload the revised manuscript. Therefore, we outline the changes we have made as indicated on the updated version of the manuscript showing marked changes (in red) as an attachment here.

We hope you find these changes agreeable and we look forwards to hearing from you in the future.

Yours sincerely and on behalf of co-authors,

Jason Hilton

**Reviewer #2:**

Overall, reviewer 2 makes some detailed comments on the geological context of the study but nothing that impacts the results or conclusions presented. We will reply to their comments individually below.

1. The paleogeographic context is inadequate. Fig. 1a is a cartoon with virtually no documentation of how the North China Block was positioned in the Early Permian, e.g., Liu1990 cited is a one-page comment/reply and Blakely2011 has no quantitative analysis for determination of paleolatitude, whereas Yang+Lei1987 and Zhou2002 are not readily accessible and would need to be described here for a broader audience. The authors state that the NCP was located at ~5-15°N but there could be a huge lithostratigraphic consequence between being in the tropical humid belt (5°) and the arid belt (15°). Unless the authors have alternative methods, paleolatitudes are based on paleomagnetism and hence the authors might consult references to the modern paleomagnetic literature and syntheses for the NCB and Tethyan environs in papers like Torsvik+2012 ESR or Kent+Muttoni2020 Palaeo3.

**Response:** We disagree with the overall sentiment here – the information is provided is coparable to other papers on the topic utilizing similar appraches, but if the reviewer wants us to provide additional context, we can do that. We note this does not change the results or conclusions presented.

We have reviewed palaeomagnetic data from the Dafengkou section in the Yuzhou coalfield from the early to late Early Permian (including the Taiyuan, Shanxi, and Shangshihezi

formations). These data show that from the early Early Permian to late Early Permian, the palaeolatitude change in the study area is between 11.0°N and 11.4°N, that is, it is within the equatorial humid climate zone (Zhu et al., 1996).

| Period           | sampling
point | mean direction of magnetization |       |       | α95  | stability | Palaeomagnetic pole position |            |      |      | Palaeolatitude |
|------------------|-------------------|---------------------------------|-------|-------|------|-----------|------------------------------|------------|------|------|----------------|
|                  |                   | D (°)                           | l (°) | K (°) |      | valuation | Plat.(°N)                    | Plong.(°N) | Dp   | Dm   | position       |
| P 2 1 | 6                 | 143.2                           | -21.2 | 12.9  | 19.4 | R         | -49.2                        | 177.4      | 10.7 | 20.4 | 11.0           |
| P 1 1 | 3                 | 124.9                           | -22.0 | 82.1  | 13.7 | R         | -35.1                        | 192.5      | 7.7  | 14.5 | 11.4           |
| P 1   | 9                 | 136.9                           | -21.7 | 16.1  | 13.2 |           | -44.6                        | 183.4      | 7.4  | 14.0 | 11.2           |

Table 1 The Characteristic remanent magnetization directions, palaeomagnetic pole, and palaeolatitude of Permian ofDafengkou profile in Yuzhou, Henan Province

According to palaeomagnetic data from the Dafengkou section in the Yuzhou Coalfield during the early Early Permian (P1) to late Early Permian (P21) (Table 1; see below), we determined the exact location of the study area through time and have modified this in the geological background section. We have revised figure 1, and replaced "Blake's paleogeographic map (Fig. 1a)" with the "generalized tectonic map of present-day China", and also replaced the "Palaeofacies map of the NCP during the Cisuralian (modified from Liu, 1990) (Fig. 1b)" with the "Simplified tectonic map of the present-day NCP (modified from Liu et al., 2013)". In these maps, we have added modern latitude and longitude (Figs. 1a, 1b; see below). However, this information does not affect the geological background in our manuscript. We have added a map of the the facies and palaeogeographic evolution of the NCP over time from the late Bashkirian to the Wordian (~318 Ma – 265.1 Ma) in the revised manuscript (Figure 9) to provde full context on this.